# Prognostic Impact of Non-Cardiac Comorbidities on Long-Term Prognosis in Patients with Reduced and Preserved Ejection Fraction following Acute Myocardial Infarction

**DOI:** 10.3390/jpm13071110

**Published:** 2023-07-08

**Authors:** Lidija Savic, Igor Mrdovic, Milika Asanin, Sanja Stankovic, Ratko Lasica, Dragan Matic, Damjan Simic, Gordana Krljanac

**Affiliations:** 1Faculty of Medicine, University of Belgrade, 11000 Belgrade, Serbia; igormrd@gmail.com (I.M.); masanin2013@gmail.com (M.A.); drlasica@gmail.com (R.L.); dragan4m@gmail.com (D.M.); gkrljanac@gmail.com (G.K.); 2University Clinical Center of Serbia, Emergency Hospital, Coronary Care Unit & Cardiology Clinic, 11000 Belgrade, Serbia; simicdamjan@hotmail.com; 3Center for Medical Biochemistry, Emergency Hospital, University Clinical Center of Serbia, 11000 Belgrade, Serbia

**Keywords:** non-cardiac comorbidities, ejection fraction, myocardial infarction, prognosis

## Abstract

Background: We aimed to analyze the prevalence and long-term prognostic impact of non-cardiac comorbidities in patients with reduced and preserved left-ventricular ejection fraction (EF) following ST-elevation myocardial infarction (STEMI). Method: A total of 3033 STEMI patients undergoing primary percutaneous coronary intervention (pPCI) were divided in two groups: reduced EF < 50% and preserved EF ≥ 50%. The follow-up period was 8 years. Results: Preserved EF was present in 1726 (55.4%) patients and reduced EF was present in 1389 (44.5%) patients. Non-cardiac comorbidities were more frequent in patients with reduced EF compared with patients with preserved EF (38.9% vs. 27.4%, respectively, *p* < 0.001). Lethal outcome was registered in 240 (17.2%) patients with reduced EF and in 40 (2.3%) patients with preserved EF, *p* < 0.001. Diabetes and chronic kidney disease (CKD) were independent predictors for 8-year mortality in patients with preserved EF. In patients with reduced EF, CKD was independently associated with 8-year mortality. Conclusion: In patients who had reduced EF, the prevalence of non-cardiac comorbidities was higher than in patients who had preserved EF after STEMI. Only diabetes mellitus and CKD were independently associated with 8-year mortality in analyzed patients.

## 1. Introduction

The introduction and refinement of primary percutaneous coronary intervention (pPCI) in the treatment of patients with acute myocardial infarction with ST segment elevation (STEMI) has significantly improved survival and reduced the occurrence of complications, which is why patients with stable chronic coronary disease post-STEMI represent an increasingly important and prevalent population [1,2,3,4]. Non-cardiac comorbidities are frequent in patients with acute myocardial infarction (AMI) [4,5]. The presence of non-cardiac comorbidities in patients with AMI can be explained by the presence of the same risk factors for both AMI and non-cardiac comorbidities (e.g., smoking is also a risk factor for chronic obstructive pulmonary disease, peripheral artery disease, stroke, etc.), while some non-cardiac comorbidities are risk factors for coronary disease (e.g., diabetes mellitus, hypertension, chronic kidney disease, etc.) [4]. The results of studies analyzing the prognostic significance of non-cardiac or non-cardiovascular comorbidities in patients with STEMI and/or non-STEMI have shown that the presence of at least one non-cardiac comorbidity has a negative impact on short-term and long-term prognosis [2,3,4,5,6,7,8,9,10,11].

Left-ventricular ejection fraction (EF) is considered one of the most important predictors of prognosis after STEMI. The percentage of patients with reduced left-ventricular systolic function (i.e., reduced EF) is smaller in the pPCI era than in the thrombolytic era because establishing normal blood flow through the infarcted artery leads to a reduction in the myocardial necrotic zone [12]. It can be said that in the era of primary PCI, there is a higher percentage of patients with preserved systolic function than patients with reduced systolic function of the left ventricle. Patients with preserved EF generally have an excellent short- and long-term prognosis [1]. Predictors of long-term prognosis, as well as the influence of non-cardiac comorbidities, may be different in patients with reduced and preserved left-ventricular systolic function. To the best of our knowledge, the influence of non-cardiac comorbidities on the long-term prognosis of patients in relation to the EF value after STEMI has not been analyzed as of yet.

The aim of this study was to analyze the prevalence and long-term prognostic impact of non-cardiac comorbidities in patients with reduced and preserved EF following ST-elevation myocardial infarction. 

## 2. Method

### 2.1. Study Population, Data Collection and Definitions

The present study enrolled 3115 consecutive patients hospitalized between December 2005 and January 2012, who were included in the prospective University Clinical Center of Serbia STEMI Register. The purpose of the prospective University Clinical Center of Serbia STEMI Register has been published elsewhere [13].

In brief, the objective of the register is to gather complete and representative data on the management and short- and long-term outcomes of patients with STEMI, undergoing primary PCI in the center. All consecutive patients with STEMI, aged 18 or older, who were admitted to the coronary care unit after undergoing pPCI in the center, were included in the register. All patients for whom data were entered into the register received written information regarding their participation in the register and the long-term follow-up, and their verbal consent for enrollment was obtained. Patients with cardiogenic shock at admission were excluded. The flowchart of patient selection is shown in Figure 1.

Coronary angiography was performed via the femoral approach. Primary PCI and stenting of the infarct-related artery (IRA) was performed according to the standard technique. Aspirin (300 mg) and clopidogrel (600 mg) were administered to all eligible patients before pPCI. Selected patients, with visible intracoronary thrombi, were also given the GP IIb/IIIa receptor inhibitor during pPCI. Flow grades were assessed according to the Thrombolysis in Myocardial Infarction (TIMI) criteria. After pPCI, patients were treated according to current guidelines.

Demographic, baseline clinical, angiographic, and procedural data were collected and analyzed. Echocardiographic examination was performed within the first three days after pPCI. The left-ventricular ejection fraction (EF) was assessed according to the biplane method in classical two- and four-chamber apical projections. Patients were divided into two groups: patients with preserved EF (EF ≥ 50%) and patients with reduced EF (EF < 50%). 

Non-cardiac comorbidities were defined by means of hospital diagnoses or with the help of retrieved prescriptions that were issued before hospital admission (i.e., before the index event—STEMI). We considered the non-cardiac comorbidities included in the Charlson Comorbidity Index—CCI [14]. On the basis of the CCI, we analyzed the following non-cardiac comorbidities: previous stroke; peripheral artery disease (PAD); chronic obstructive pulmonary disease (COPD); diabetes mellitus (DM); chronic kidney disease (CKD); the value of the estimated glomerular filtration rate at admission < 60 mL/min/m^2^, using the Modification of Diet in Renal Disease equation; carcinoma; peptic ulcer disease; psychiatric disorders; and liver disease. We also analyzed anemia (according to the World Health Organization criteria baseline, defined as a hemoglobin level < 130 g/L in men, and a hemoglobin level < 120 g/L in women) and obesity (defined as body mass index ≥ 30 kg/m^2^). Anemia was included in the analysis due to its relatively frequent presence in our patients, its importance in the application and duration of application of dual antiplatelet therapy, and the known negative impact on the prognosis of patients with acute coronary syndrome [15].

Patients were followed-up with eight years after enrolment. Follow-up data were obtained by scheduled telephone interviews and outpatient visits. We analyzed all-cause mortality.

### 2.2. Ethics

The study protocol was approved by the ethics committee of the University of Belgrade, Faculty of Medicine (approval number 470/II-4, 21 February 2008). The study was conducted in accordance with the principles set forth in the Helsinki Declaration. Written informed consent was obtained from all patients for their participation in the register.

### 2.3. Statistical Analysis

Categorical variables were expressed as frequency and percentage, while the continuous variables were expressed as the median (med), with 25th and 75th quartiles (IQR). Analysis for normality of data was performed using the Kolmogorov–Smirnov test. Baseline differences between groups were analyzed using the Mann–Whitney test for continuous variables, and the Pearson *X*^2^ test for categorical variables. The Kaplan–Meier method was used to construct the probability curves for eight-year survival, while the difference between patients with and without non-cardiac comorbidities was tested with the log-rank test. Multiple cox analysis (backward method, with *p* < 0.10 for entrance into the model) was used to identify independent risk factors for the occurrence of eight-year all-cause mortality. We included all variables that differed in the preliminary analysis between patients who were alive and patients who died during follow-up. Two-tailed *p* values of <0.05 were considered to indicate a statistically significant difference. The SPSS statistical software, version 19, was applied (SPSS Inc, Chicago, IL, USA).

## 3. Results

Out of the 3115 patients analyzed, 1389 (44.5%) patients had reduced EF, and 1726 (55.4%) patients had preserved EF. Non-cardiac comorbidities at admission were present in a total of 1014 (32.5%) patients. Amongst the patients with non-cardiac comorbidities, 565 (18.1%) patients had one non-cardiac comorbidity, while 449 (14.4%) patients had two or more non-cardiac comorbidities. The most prevalent non-cardiac comorbidity was DM, followed by CKD, obesity, anemia, COPD, previous stroke, and PAD. The prevalence of other comorbidities was less than 0.1%. Compared to patients with preserved EF, patients with reduced EF were older, they had heart failure at admission more frequently, as well as previous hypertension, multivessel coronary artery disease at initial angiogram, and post-procedural flow TIMI < 3. Non-cardiac comorbidities at hospital admission were more frequently present in patients with reduced EF compared to patients with preserved EF. Baseline characteristics; laboratory, angiographic, and procedural characteristics, therapy at discharge, and comorbidities in the whole cohort and in patients with reduced and preserved EF are shown in Table 1.

During the 8-year follow-up, the lethal outcome (all-cause mortality) was registered in a total of 244 (7.8%) patients; in the group with preserved EF, the lethal outcome was registered in 40 (2.3%) patients; and in the group with reduced EF, the lethal outcome was registered in 240 (17.2%) patients, *p* < 0.001. Causes of mortality in all of the analyzed patients were predominantly cardiovascular (N = 228, 93.3% of all deaths). Cardiovascular causes included fatal re-infarction, progression of heart failure, sudden death, and stroke. Non-cardiovascular causes of death (such as cancer, ileus, pneumonia) were registered in 16 patients (6.5% of all deaths).

Patients with non-cardiac comorbidities had a higher 8-year all-cause mortality compared to patients without non-cardiac comorbidities, and mortality increased if two or more comorbidities were present at the same time, as shown in Figure 2.

Predictors for the occurrence of all-cause mortality during 8-year patient follow-up are presented in Table 2.

## 4. Discussion

The results of our study showed that non-cardiac comorbidities were present in about a third of the analyzed patients. In patients who had reduced EF after STEMI, the prevalence of non-cardiac comorbidities was higher compared to patients who had preserved EF after STEMI. Regardless of the EF value, 8-year mortality was higher in patients with the presence of at least one non-cardiac comorbidity compared to patients without any non-cardiac comorbidities. In a multivariate analysis, we found that diabetes mellitus and chronic kidney disease were independent predictors of 8-year all-cause mortality in patients with preserved EF, with the effect of DM being significantly greater than the prognostic impact of CKD. In patients with reduced EF, CKD was a strong independent predictor and the only comorbidity that had an independent prognostic impact on 8-year all-cause mortality.

### 4.1. Non-Cardiac Comorbidities in Patients with Myocardial Infarction

The prevalence of non-cardiac comorbidities in our patients is mostly in accordance with the data from the literature, bearing in mind, however, the data found in the literature depend on the design of the study, the population of analyzed patients, and the selected comorbidities for analysis [2,7,10,16]. In a systematic review by Breen et al., it was found that the prevalence of non-cardiac comorbidities in different studies ranged from 33% to 69%, with a mean of 56% [16]. Despite the high prevalence of multimorbidity in the ACS population, there was a lack of consistency in the way multimorbidity was measured and characterized. Currently, there is no gold standard for comorbidity measuring [16].

The lack of studies analyzing the prognostic impact of non-cardiac comorbidities in patients with STEMI treated with pPCI (as well as in relation to left-ventricular systolic function) limits direct comparison of our results with data from the literature. On the other hand, the unfavorable impact of non-cardiac comorbidities on the prognosis of patients with AMI (STEMI and non-STEMI) has been analyzed in numerous studies, bearing in mind, however, that the prognostic impact was analyzed in a shorter follow-up compared to our study [2,4,8,9,16,17,18,19]. All studies show that the presence of a greater number of comorbidities is associated with higher mortality or more frequent occurrence of other complications. Furthermore, in almost all studies, there is a significant prognostic impact of DM and/or CKD on the patient’s prognosis.

A review by Rashid et al. showed that an increase in the value of the Charlson Comorbidity Index (CCI) increases the mortality of patients with acute coronary syndrome by 30% and the mortality of patients treated with primary PCI by 20%. This review also showed that a CCI score value of above 2 increases the risk of mortality by 2.5 times in patients with acute coronary syndrome, and by about 3 times in patients undergoing PCI [17]. In two Swiss cohort studies of patients with acute coronary syndrome, 8330 patients with ST-segment elevation myocardial infarction (STEMI) undergoing percutaneous coronary intervention were analyzed, during the period 2005–2012 [18]. It was shown that CCI scores ≥ 2 increased the one-year risk of major adverse cardiovascular and cerebrovascular events by 40% [18]. In a nationwide cohort study by Schmidt et al., the impact of comorbidities on the prognosis of patients with AMI was analyzed, as well as the interaction between comorbidities and AMI. It was shown that the interaction effect was the greatest within the first 30 days of follow-up, but similar dose–response patterns were also observed between 31 and 365 days of follow-up, as well as during 1 to 5 years of follow-up [9]. In the study by Rapsomaniki et al., mortality and the occurrence of bleeding in patients with AMI after one year were analyzed. In this study, a strong predictor of three-year all-cause death was CKD, along with heart failure, COPD, and a history of cancer. Unlike our study, the aforementioned studies analyzed patients with STEMI and NSTEMI, and revascularization was not performed in all STEMI patients [8]. A systematic review by Johanson et al. included studies analyzing all-cause death up to 5 years after AMI. Compared with the general population, MI survivors remain at high risk, particularly older patients and those with comorbidities, such as hypertension, DM, CKD, PAD, or history of stroke [11].

### 4.2. Non-Cardiac Comorbidities in Relation to EF

The prognostic impact of comorbidities in relation to the EF value in patients with heart failure has been analyzed in the literature. In a study by Iorio et al., a similar prevalence and similar negative prognostic impact of non-cardiac comorbidities was found in patients with heart failure and reduced EF (HFrEF), as well as in patients with heart failure and preserved EF (HFpEF). In this study, CKD, anemia, COPD, and PAD had the strongest and most significant association with mortality [20]. In a study by van Deursen et al., comorbidities in patients with chronic heart failure were analyzed and it was found that CKD, anemia, and diabetes mellitus were independent predictors of mortality and heart failure hospitalization [21]. CKD showed the greatest prognostic significance, i.e., 41%, of all-cause mortality in the analyzed population was attributable to CKD comorbidity [21]. Additionally, in a study by Yang et al., it was found that the association between DM and 5-year all-cause mortality in patients with heart failure and reduced EF existed only if CKD was also present [22]. These results in the aforementioned studies are in concordance with our results.

### 4.3. Possible Mechanisms of the Negative Impact of Comorbidities on the Prognosis of Patients with AMI

Non-cardiac comorbidities may be risk factors for coronary disease, which is why secondary prevention is extremely important in these patients. This particularly applies to the coexistence of DM and CKD [22]. There are also multiple pathophysiological mechanisms that can account for the negative prognostic impact of impaired renal function in patients with acute myocardial infarction and/or heart failure. Complications of advanced CKD, such as hypercalcemia, anemia, and disorders of the blood coagulation system, increase the risk of atherosclerotic disease progression when they co-occur with other risk factors, especially DM. Sympathetic and numerous neurohormonal mechanisms, inflammation, free radicals, and other factors can significantly influence the development and progression of cardiorenal or renocardial syndrome [23,24]. CKD patents exhibit a high incidence of cardiovascular complications that are characterized by complex alterations in the mechanical and electrical properties of the heart [25]. Furthermore, the presence of non-cardiac comorbidities may reduce the possibility of taking the guideline-recommended treatment after AMI [4,9,10]. There is also a known treatment–risk paradox with regard to the provision of invasive coronary procedures. It is known that, in patients with ACS and one or more comorbidities, revascularization procedures are generally performed less often [10,16]. In our study, this was not the case, because all our patients were treated with pPCI. In the study by Yadegarfar et al., it was found that the presence of each analyzed comorbidity, with the exception of hypertension, renal failure, and PAD, was associated with a reduced chance of receiving optimal care [10]. In some situations, comorbidities may reduce the effectiveness of optimal or guideline-recommended treatment in patients with AMI [10]. Furthermore, rehabilitation after AMI is usually performed less often in patients with comorbidities [16], which can affect prognosis, as well as the quality of life.

### 4.4. Clinical Significance of Our Study

We feel that our findings add to the existing knowledge on the prevalence of non-cardiac comorbidities among unselected STEMI patients and their influence of mortality in the primary PCI era. The frequent presence of non-cardiac comorbidities in patients with STEMI represents a significant challenge in the treatment of these patients, both during hospitalization and upon discharge from the hospital [26,27]. Multidisciplinary efforts to better manage cardiovascular risk factors may have an additional secondary preventive role, since the most common causes of death in our patients were cardiovascular [5]. Even in patients who have preserved EF, who are without complications after STEMI, and have CKD and/or DM, prolonged dual antiplatelet therapy is considered, taking into account the hemorrhagic risk. Attention should also be focused on therapy for non-cardiac comorbidities due to possible drug interactions, but also the possible effects on cardiac function and the occurrence of adverse events after AMI [8,10,16].

#### Study Limitations

A number of limitations to our study warrant being mentioned. The study is unicentric and observational, but it is controlled, prospective, and has included consecutive patients, thus limiting possible selection bias. Patients enrolled in the study were hospitalized between 2006 and 2012. We did not use other measures for determining systolic function, such as myocardial deformation imaging. However, many cornerstone clinical trials conducted so far have used EF to stratify patients [28,29]. The longitudinal measurement of EF was not performed for the purpose of evaluating the improvement in left-ventricular function after STEMI [30]. Some comorbidities were present in less than 0.1% of our patients, so their prognostic impact cannot be ruled out with certainty. We did not analyze the impact of comorbidities that develop during follow-up (e.g., after STEMI). Patients were treated with clopidogrel as there were no patients treated with more recently developed antithrombotic drugs (ticagrelor was not available for routine administration to patients at the time of their entry into the register), which could have influenced the prognosis of the patients. The study was not designed to evaluate whether changing pharmacological treatment would have an impact on the long-term outcome in the analyzed patients.

## 5. Conclusions

About a third of the analyzed patients with STEMI had at least one non-cardiac comorbidity. Patients with reduced, as well as those with preserved, EF after STEMI and the presence of at least one non-cardiac comorbidity had a higher 8-year all-cause mortality compared to patients without non-cardiac comorbidities. However, only diabetes mellitus and chronic kidney disease had an independent prognostic impact on 8-year all-cause mortality in patients with preserved EF, while in patients with reduced EF, CKD was the only non-cardiac comorbidity that was an independent predictor of 8-year all-cause mortality.

## Figures and Tables

**Figure 1 jpm-13-01110-f001:**
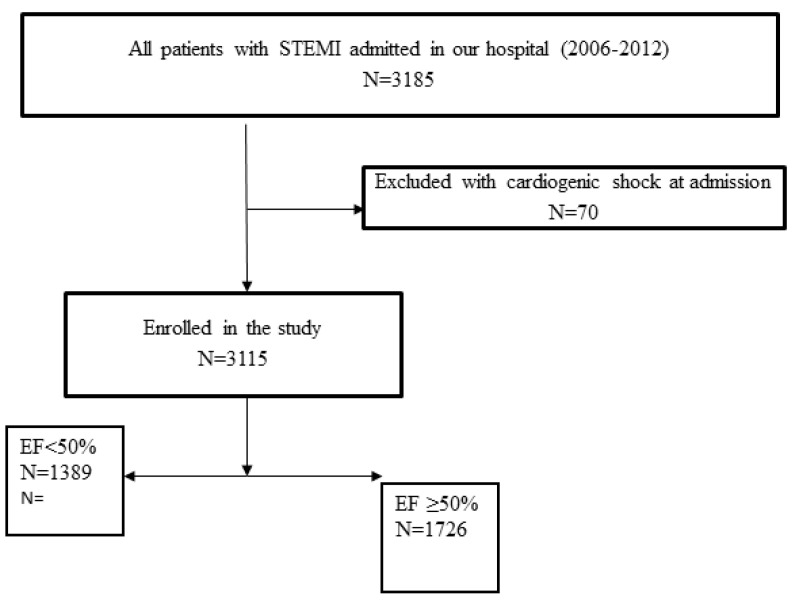
Flow-chart of the patient selection.

**Figure 2 jpm-13-01110-f002:**
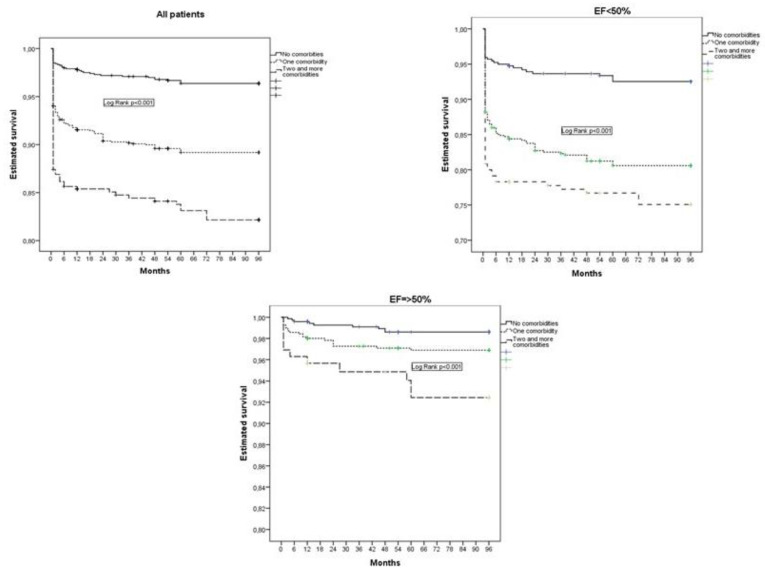
Estimated survival according to the presence of non-cardiac comorbidities (Kaplan–Meier curves).

**Table 1 jpm-13-01110-t001:** Baseline clinical, laboratory, angiographic, and procedural characteristics and therapy at discharge of the study patients.

Characteristics	All Patients *N* = 3115	Reduced EF *N* = 1389	Preserved EF *N* = 1726	*p* Value (Reduced vs. Preserved EF)
Age, years med(IQR)	60 (52, 59)	61 (54, 71)	57 (50, 67.5)	<0.001
Female, *n* (%)	867 (27.8)	407 (29.3)	460 (26.7)	0.160
Previous MI, *n* (%)	327 (10.5)	193 (13.9)	134 (7.8)	<0.001
Previous PCI, *n* (%)	84 (2.7)	57 (4.1)	27 (1.6)	<0.001
Previous CABG, *n* (%)	52 (1.7)	28 (2)	24 (1.4)	0.176
Hypertension, *n* (%)	2089 (67.1)	967 (69.1)	1122 (62.5)	0.012
HLP, *n* (%)	1890 (60.7)	814 (58.7)	1076 (62.3)	0.034
Smoking, *n* (%)	1656 (53.2)	648 (46.7)	1008 (58.4)	<0.001
Pain duration, hours med(IQR)	2.5 (1.5, 4)	3 (1.5, 4)	2.5 (1.5, 4)	0.091
Atrial fibrillation on initial ECG, *n* (%)	215 (6.9)	160 (11.5)	55 (3.2)	<0.001
Complete AV block, *n* (%)	144 (4.6)	75 (5.4)	68 (3.9)	0.081
Killip class > 1, *n* (%)	390 (12.5)	349 (25.1)	41 (2.4)	<0.001
Systolic BP at admission, med(IQR)	135 (120, 150)	130 (120, 150)	140 (120, 150)	<0.001
Heart rate at admission med(IQR)	78 (70, 90)	80 (70, 96)	75 (66, 82)	0.005
BBB on initial ECG, *n* (%)	119 (3.8)	85 (6.2)	134 (7.8)	<0.001
Multivessel disease, *n* (%)	1763 (56.6)	868 (62.5)	895 (51.9)	<0.001
Three-vessel disease, *n* (%)	837 (27.9)	436 (31.4)	401 (23.2)	<0.001
LM stenosis, *n* (%)	191 (6.1)	108 (7.8)	83 (4.8)	<0.001
Pre-procedural flow TIMI 0, *n* (%)	2148 (69)	1031 (74.2)	1113 (64.8)	<0.001
Post-procedural flow TIMI < 3, *n* (%)	146 (4.7)	109 (7.8)	37 (2.1)	<0.001
CK MB, med (IQR)	1869 (986, 3475)	2671 (1333, 4671)	1529 (877, 2779)	<0.001
eGFR, med (IQR)	90.3 (69.6, 110.6)	83.2 (64.1, 104.5)	93.6 (75.1, 114)	<0.001
EF, med(IQR)	50 (40, 55)	40 (35, 45)	55 (50, 58)	<0.001
Non-cardiac comorbidities, *n* (%)	1014 (32.5)	541 (38.9)	473 (27.4)	<0.001
One comorbidity, *n* (%)	565 (18.1)	279 (20)	286 (16.6)	<0.001
Two or more comorbidities, *n* (%)	449 (14.4)	262 (18.9)	187 (10.9)	<0.001
Diabetes, *n* (%)	610 (19.6)	330 (23.7)	280 (16.2)	<0.001
CKD, *n* (%)	489 (15.7)	292 (21)	197 (11.4)	<0.001
Obesity, *n* (%)	490 (15.7)	206 (14.9)	284 (16.3)	0.047
Anaemia, *n* (%)	250 (8)	138 (9.1)	112 (6.5)	<0.001
Previous stroke, *n* (%)	126 (4)	71 (5.1)	55 (3.2)	0.007
COPD, *n* (%)	40 (1.3)	12 (0.8)	28 (1.6)	<0.001
PAD, *n* (%)	29 (0.9)	18 (1.3)	11 (0.6)	0.469
Peptic ulcer disease, *n* (%)	14 (0.4)	8 (0.5)	6 (0.3)	<0.001
Liver disease, *n* (%)	3 (0.09)	0	3 (0.2)	<0.001
Psychiatric disorder, *n* (%)	2 (0.09)	1 (0.07)	1 (0.05)	0.123
Carcinoma, *n* (%)	9 (0.2)	5 (0.3)	4 (0.2)	0.154
Therapy at discharge *				
Beta blockers, *n* (%)	2987 (85.5)	1211 (95.9)	1647 (95.8)	0.895
ACE inhibitors, *n* (%)	2784 (89.4)	1034 (74.4)	1370 (79.4)	0.001
Statin, *n* (%)	3033 (97.4)	1300 (93.6)	1725 (99.9)	0.004
Diuretic, *n* (%)	478 (15.3)	385 (27.2)	93 (5.4)	<0.001
Amiodarone, *n* (%)	82 (2.6)	42 (3.1)	40 (2.4)	0.211

* All patients were on aspirin and clopidogrel. med = median; IQR = interquartile range; EF = left-ventricular ejection fraction; MI = myocardial infarction; CABG = coronary aortocoronary bypass grafting; PCI = percutaneous coronary intervention; HLP = hyperlipidemia; AV = atrioventricular; HF = heart failure; BP = arterial blood pressure; BBB = bundle branch block on ECG at admission; CK-MB = creatine kinase MB isoform; eGFR = estimated glomerular filtration rate; PAD = peripheral artery disease; CKD = chronic kidney disease; COPD = chronic obstructive pulmonary disease.

**Table 2 jpm-13-01110-t002:** Predictors for 8-year mortality (Cox regression model).

	Univariable Analysis		Multivariable Analysis	
	HR (95%CI)	*p* Value	HR (95%CI)	*p* Value
All Patients				
Age (years)	1.04 (1.02–1.05)	<0.001	1.04 (1.02–1.05)	<0.001
Previous infarction	1.07 (0.83–1.55)	0.730		
Systolic blood pressure at admission mmHg	0.99 (0.99–1.00)	0.190		
Heart rate at admission	1.0 (0.99–1.0)	0.691		
CK max	1.00 (1.00–1.01)	0.802		
EF < 50%	3.66 (2.49–5.41)	<0.001	2.28 (2.28–4.07)	<0.001
Killip > 1 at admission	2.59 (1.81–3.29)	<0.001	2.14 (1.51–3.05)	<0.001
Pre-procedural TIMI 0	1.25 (0.79–1.95)	0.336		
Post-procedural TIMI < 3	2.30 (1.58–3.32)	<0.001	2.0 (1.10–3.66)	0.023
Acute bundle branch block	1.62 (1.15–2.20)	0.020		
Atrial fibrillation on initial ECG	1.55 (1.12–1.99)	0.032		
Complete AV block at admission	2.07 (1.06–4.24)	0.033	1.97 (1.06–2.08)	0.042
Three-vessel disease	1.51 (1.14–1.99)	0.040		
Diabetes mellitus	2.08 (1.30–2.78)	0.010	1.79 (1.18–2.48)	0.015
CKD	2.68 (1.37–3.68)	<0.001	1.69 (1.17–2.68)	0.010
Anemia	1.42 (1.12–1.98)	0.054		
Preserved EF ≥ 50%				
Age (years)	1.03 (1.01–1.07)	0.050	1.04 (1.01–1.08)	0.040
Systolic blood pressure at admission mmHg	1.0 (0.98–1.01)	0.721		
Heart rate at admission	1.01 (0.98–1.04)	0.091		
CK max	1.0 (1.0–1.01)	0.779		
Killip > 1 at admission	1.32 (1.08–3.65)	0.053		
Three-vessel disease	1.25 (0.69–2.64)	0.071		
Pre-procedural TIMI 0	0.89 (0.85–1.28)	0.861		
Post-procedural TIMI < 3	2.91 (0.99–12.3)	0.055		
Diabetes mellitus	2.24 (1.11–4.64)	0.027	1.93 (1.21–3.75)	0.032
CKD	1.56 (1.28–2.24)	0.017	1.55 (1.07–2.23)	0.047
Anemia	1.39 (1.02–2.89)	0.051		
Reduced EF < 50%				
Age (years)	1.04 (1.02–1.05)	<0.001	1.03 (1.02–1.05)	<0.001
Previous infarction	1.59 (1.11–1.97)	0.072		
Systolic blood pressure at admission, mmHg	0.99 (0.98–0.99)	0.056		
Heart rate at admission	1.0 (0.99–1.0)	0.131		
CK max	1.0 (0.99–1.01)	0.269		
Pre-procedural TIMI = 0	1.23 (0.78–2.08)	0.423		
Post-procedural TIMI < 3	4.42 (3.12–6.09)	<0.001	2.49 (1.80–3.28)	<0.001
Killip > 1 at admission	3.68 (2.81–4.23)	<0.001	2.40 (1.90–3.74)	<0.001
Atrial fibrillation on initial ECG	1.49 (1.11–2.09)	0.030		
Acute bundle branch block	1.89 (1.28–2.38)	0.020		
Three-vessel disease	1.51 (1.11–2.04)	0.008	1.71 (1.30–2.18)	0.035
Diabetes mellitus	1.69 (1.22–2.19)	0.001		
CKD	3.20 (2.14–4.35)	<0.001	2.09 (1.18–2.52)	0.030
Anemia	2.23 (1.56–3.20)	<0.001		

EF = left-ventricular ejection fraction; TIMI = thrombolysis in myocardial infarction; CKD = chronic kidney disease; PAD = peripheral artery disease; TIMI = thrombolysis in myocardial infarction; CK = creatine kinase; AF = atrial fibrillation; AV = atrioventricular.

## Data Availability

The data presented in this study are available on request from the corresponding author. The data are not publicly available due to privacy and ethical reasons.

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
