# Peer review of "Prognostic Impact of Non-Cardiac Comorbidities on Long-Term Prognosis in Patients with Reduced and Preserved Ejection Fraction following Acute Myocardial Infarction"

_jpm, 2023, doi:10.3390/jpm13071110_

Round 1

Reviewer 1 Report

This article attempts to assess the impact of comorbidity on outcomes in patients with STEMI, depending on the LV EF. However, the novelty of this study raises a question. Because there are already a number of studies indicating that CKD affects mortality and prognosis in patients with reduced LVEF (for example, https://www.ncbi.nlm.nih.gov/pmc/articles/PMC3904800/). There is also enough information about the impact of diabetes mellitus on prognosis after MI.

In addition, there are a number of comments on the text:

1. In the abstract, the conclusion contradicts the results.

2. When using references to literary sources, it is recommended to replace [2, 3, 4, 5, 6, 7, 8, 9, 10, 11] with [2-11].

3. It is recommended to pay attention to the use of abbreviations and transcripts in the text. PCI, CKD, DM are not decoded in the abstract. The text gives the transcript EF twice.

4. In table 1, there is no decoding of abbreviations at all. Table 2 does not contain all transcripts.

5. From the list of references, 15 out of 27 sources are older than 5 years ago.

6. The suggestion “Obesity was analyzed because it represents a global health problem” is recommended to be removed from the materials and methods section.

It is recommended to modify the sentence “Due to improved survival, patients with stable chronic coronary disease post STEMI represent an increasingly important and prevalent population.”

Introduction, materials and methods, the results are well written, the discussion does not reflect the discussion of the results obtained. The novelty of the study and the conclusion are questionable.

Reviewer 2 Report

In this paper, authors aim to determine comorbidities and how they relate to the progression of cardiac outcomes after STEMI. They use a dataset involving 3,115 patients throughout the 8 year window. They determine the significant comorbidities in both reduced EF and preserved EF patient groups to conclude that CKD might be a major determinant of cardiac outcomes for both groups.

Major comments:

Are the comorbidities indicated in the survival curves non-cardiac or cardiac comorbidities? If the authors used only non-cardiac comorbidities, they should specify it in the figure legend and in the text.

Authors should specify which figure/graph they are referring to in the text. The claims such as

"Patients with non-cardiac comorbidities had a higher 8-year all-cause mortality, as
compared to patients without non-cardiac comorbidities, and mortality increased if two
or more comorbidities were present at the same time."

should be followed up by the graph the authors are referring to. (Figure 2, top graph/or letter) etc.

Please indicate p values and values for multivariable analysis for all comorbidities even though they are not significant.

Were the significant comorbidities indicated in Table 2 taken from patients with only one comorbidity or more than one? It would be preferred if authors can run multiple analyses to see which predictor has the most impact when it is the only comorbidity in a patient.

Authors should discuss mechanisms in detail in the "Possible mechanisms of the negative impact of comorbidities on the prognosis of patients with AMI". As it is right now authors does not discuss any of the mechanisms of the comorbidities that they claim to have an effect on cardiac outcomes. Please discuss further in detail why comorbidities such as Diabetes or CKD can exacerbate cardiac disease and cause poor prognosis.

Authors should discuss research papers involving CKD and cardiac diseases that actually investigated causality and interdependence between these two conditions (e.g PMID 32064746, 31709320)

Minor comments:

Please correct "Man-Whitney" in the Statistical Analysis section with "Mann-Whitney".

Reviewer 3 Report

This study reports the role of non-cardiovascular comorbidities in predicting long-term prognosis in a cohort of patients with STEMI. Whereas the prognostic significance of non-cardiac comorbidities has been well known in heart failure patients, this paper described the long-term impact in STEMI patients, which should be interesting to JPM readership. However, this reviewer has a few concerns on this work.

1.      Why the authors did not include in multivariate analysis of cardiac biomarkers as surrogate of the infarct size, which has long been known as a key variable for prognosis of this cohort of patients. Whereas CK was determined, how about Troponin-I?

2.      for patients with MI, it may not be appropriate to assign as “preserved EF” because all patients would be expected to exhibit reduction in EF but varying in the degree. Indeed, in your “preserved EF” group, the mean EF was 55%, lower than the normal range.

3.      Please clarify as whether comorbidities studied occurred only prior to MI? how about the significance of comorbidities that developed post MI?

4.      Table: The P-values of a number of parameters need double-checked for the accuracy.

5.      Discussion part: several paragraphs in this section read like a review article without referring findings from own study. Also, this section can be shortened.

language polishing is required and this paper can also be shortened in particular Discussion section.

Round 2

Reviewer 1 Report

Indeed, the authors have done a lot of work on the bugs.

They tried to take into account all the comments, but the novelty of the study remains controversial.

What practical application the obtained data can give is not clear.

From the list of references, 15 out of 30 sources are older than 5 years ago. 

Author Response

We would like to thank you for your review of our paper. We have done our best to answer all the questions and to make corrections in the manuscript in accordance with the reviewer’s comments

Reviewer 3 Report

No further comment to the authors, who have responded to my previous comments.

Author Response

Academic Editor's comments:

Multiple cox analysis (backward method, with p <
0.10 for entrance into the model) was used for identifying independent risk
factors for the occurrence of eight-year all-cause mortality. What risk
factors were tested? all those in table 2? authors must test for all possible
risk factors for sudden cardiac death.
It seems that only those 7 variables were tested, if so, the manuscript
cannot be accepted in this form.
if all possible confounding factors are not taken into account in table 2,
the results obtained could be distorted.
Table 2 all patients, the EF (<50% vs >=50%) must be entered as a covariate
in the model.

Answer: Thank you for comment. All variables that differed in the preliminary analysis between patients who were alive and died during follow-up were included in the Cox regression model.  We corrected Table 2 in the manuscript.
